

# Stress-induced changes in cognitive function and intestinal barrier integrity can be ameliorated by venlafaxine and synbiotic supplementations

Sarawut Lapmanee[1], Nattapon Supkamonseni[1], Sakkarin Bhubhanil[1], Nattakan Treesaksrisakul[1], Chaiyos Sirithanakorn[2], Mattaka Khongkow[3], Katawut Namdee[3], Piyaporn Surinlert[4,5], Chittipong Tipbunjong[6] and Prapimpun Wongchitrat[7]

[1] Department of Basic Medical Sciences, Faculty of Medicine, Siam University, Bangkok, Thailand
[2] Faculty of Medicine, King Mongkut's Institute of Technology Ladkrabang, Bangkok, Thailand
[3] National Nanotechnology Centre, National Science and Technology Development Agency, Pathumthani, Thailand
[4] Chulabhorn International College of Medicine, Thammasat University, Pathumthani, Thailand
[5] Thammasat University Research Unit in Synthesis and Applications of Graphene, Thammasat University, Pathumthani, Thailand
[6] Division of Health and Applied Sciences, Faculty of Science, Prince of Songkla University, Songkhla, Thailand
[7] Center for Research Innovation and Biomedical Informatics, Faculty of Medical Technology, Mahidol University, Nakhon Pathom, Thailand

Corresponding author
Prapimpun Wongchitrat,
prapimpun.won@mahidol.ac.th

## ABSTRACT

Stress profoundly impacts various aspects of both physical and psychological well-being. Our previous study demonstrated that venlafaxine (Vlx) and synbiotic (Syn) treatment attenuated learned fear-like behavior and recognition memory impairment in immobilized-stressed rats. In this study, we further investigated the physical, behavior, and cellular mechanisms underlying the effects of Syn and/or Vlx treatment on brain and intestinal functions in stressed rats. Adult male Wistar rats, aged 8 weeks old were subjected to 14 days of immobilization stress showed a decrease in body weight gain and food intake as well as an increase in water consumption, urinary corticosterone levels, and adrenal gland weight. Supplementation of Syn and/or Vlx in stressed rats resulted in mitigation of weight loss, restoration of normal food and fluid intake, and normalization of corticosterone levels. Behavioral analysis showed that treatment with Syn and/or Vlx enhanced depressive-like behaviors and improved spatial learning-memory impairment in stressed rats. Hippocampal dentate gyrus showed stress-induced neuronal cell death, which was attenuated by Syn and/or Vlx treatment. Stress-induced ileum inflammation and increased intestinal permeability were both effectively reduced by the supplementation of Syn. In addition, Syn and Vlx partly contributed to affecting the expression of the glial cell-derived neurotrophic factor in the hippocampus and intestines of stressed rats, suggesting particularly protective effects on both the gut barrier and the brain. This study highlights the intricate interplay between stress physiological responses in the brain and gut. Syn intervention alleviate stress-induced neuronal cell death and modulate depression- and memory impairment-like behaviors, and improve stress-induced gut barrier dysfunction which were similar to

those of Vlx. These findings enhance our understanding of stress-related health conditions and suggest the synbiotic intervention may be a promising approach to ameliorate deleterious effects of stress on the gut-brain axis.

## INTRODUCTION

Stress, a complex and ubiquitous element of human life, exerts diverse effects on individuals. Grasping the intricate relationship between stress, physiological reactions, and potential interventions is vital (*Kim, Pellman & Kim, 2015*; *James et al., 2023*). It remains an ongoing endeavor to uncover findings that can enhance understanding of stress-related health conditions and the need to search for effective treatments.

Stress triggers a cascade of responses within the body, often initiated by the release of stress hormones such as corticosterone in rodents and cortisol in humans (*Fowler, Bogdan & Gaffrey, 2021*). These hormones orchestrate various changes in an effort to prepare the organism to cope with stressors. The influence of stress on body weight, appetite regulation, and fluid intake particularly reflects the intricate ways in which stress can disrupt metabolic equilibrium (*Lapmanee et al., 2017*; *Francois et al., 2022*). Beyond the immediate physiological responses, stress has a profound impact on biochemical pathways. It triggers elevated urinary corticosterone levels and leads to alterations in adrenal gland weight through the activation of the hypothalamo-pituitary-adrenocortical (HPA) axis as part of the stress response (*Kamakura et al., 2016*; *Lapmanee et al., 2017*). Additionally, stress induces changes in adipose tissues affecting metabolic status, immune function, and neural health, offering insights into its broader implications (*Matsuura et al., 2015*; *Cheon, 2023*). Moreover, stress is known to increase intestinal permeability, leading to the commonly referred to condition as leaky gut (*Yu et al., 2022*). Elevated intestinal permeability, induced by corticosterone resulting from chronic or intermittent stress exposures, leads to endotoxemia due to the transport of products from the gut microbiota (*Zheng et al., 2017*; *Gurung et al., 2020*). Furthermore, this condition can exacerbate inflammation and potentially impact neural and intestinal function (*Craig et al., 2022*; *Di Vincenzo et al., 2023*). As a result, the relationship between physiological changes and the gut microbiome in response to stress exposure remains unclear.

Since the release of stress-induced glucocorticoids, these hormones can cross the blood-brain barrier, contributing to the disruption of monoamine neurotransmission homeostasis, which can lead to mood disorders and cognitive impairment (*Massart, Mongeau & Lanfumey, 2012*). The behavioral consequences of stress induction, such as depression- and memory impairment-like behaviors, serve as crucial indicators of the psychological impact of stress. Many studies commonly employ the Morris Water Maze (MWM) and the Forced Swimming Test (FST) to assess spatial memory, learning behavior, and depression-like behaviors. These tests provide valuable insights into how

stress impacts cognitive and emotional functions (*Patki et al., 2013*; *Vorhees & Williams, 2014*; *Lapmanee et al., 2017*).

Our previous findings explored the intricate effects of stress induction and potential therapeutic interventions, *i.e.*, pharmacological treatment with venlafaxine (Vlx), an antidepressant functioning as a serotonin (5-hydroxytryptamine, 5-HT)-norepinephrine (NE) reuptake inhibitor, and nutritional supplementation with synbiotics (Syn), a combination of probiotics and prebiotics. The results revealed that Vlx- and/or Syn-treated stressed rats displayed reduced anxiety-, learned fear-like behaviors, and improved object discrimination (*Lapmanee et al., 2017*, *2023a*). Importantly, these findings suggest a potential link between changes in intestinal health and hippocampal morphology. The precise cellular mechanisms underlying these observed behavioral and gastrointestinal responses continue to be the central focus of ongoing study.

While the detrimental effects of stress are apparent, there is ongoing exploration into the potential for intervention. Vlx, with its ability to modulate neurotransmitter activity by inhibiting the reuptake of 5-HT and NE, and/or Syn supplementation, nurtures the gut microbiome and promotes the release of neurotrophic factors, including brain-derived neurotrophic factor (BDNF) and glial cell-derived neurotrophic factor (GDNF), emerge as promising avenues for improvement in both animal and human studies (*Maheu et al., 2015*; *Liu, 2018*; *Haghighat, Rajabi & Mohammadshahi, 2021*). Beyond its role in stress response, GDNF has been implicated in several neuroinflammatory disorders, including Parkinson's, Alzheimer's, and diabetic neuropathies. It also plays a role in cancer, including neuroblastoma, and autoimmune diseases such as multiple sclerosis. Investigation of these intersections highlights the potential of GDNF as a versatile therapeutic target with broad implications in a variety of physiological and pathological conditions (*Liu et al., 2009*; *Straten et al., 2009*; *Fielder et al., 2018*; *Duarte Azevedo, Sander & Tenenbaum, 2020*; *Shi et al., 2021*; *Zinchuk et al., 2022*; *Singh et al., 2023*). Notably, GDNF has been shown to play a role in the pathogenesis of depressive and other psychiatric disorders (*Shen et al., 2019*). The action of Vlx in restoring the balance of 5-HT and NE neurotransmitters may aid in regulating mood and emotional responses to stress exposure by influencing the diversity of gut bacteria (*Ionescu, Rosenbaum & Alpert, 2015*; *Lapmanee et al., 2017*; *Shen et al., 2023*). Conversely, Syn, by enhancing gut health and supporting the gut-brain axis, can contribute to stress reduction through the modulation of the gut microbiota and its subsequent impact on neural function (*Dicks, 2022*; *Lapmanee et al., 2023a*).

Understanding how these interventions, specifically Vlx and Syn, influence physiological and biochemical profiles, as well as behavioral outcomes in stressed rats offers a comprehensive perspective on potential benefits. This includes the capacity to modulate intestinal permeability and promote neuroprotective mechanisms. By elucidating these intricate mechanisms, the goal is to contribute to a deeper understanding of stress-related health conditions and identify potential avenues for intervention to mitigate the impact on both the gut and the brain. Therefore, it is hypothesized that Vlx and Syn play a role in restoring gut barrier integrity and promoting the production of target neuroprotective proteins, including GDNF, in both the gut and the hippocampus-brain region regulating

memory formation and mood control. These actions, in turn, can have broader implications from animal models to humans for overall health, enhancing resilience to the effects of stress.

## MATERIALS AND METHODS

### Animals and treatments

Biological samples derived from experimental animals with all treatments were obtained in our previous study. All experimental procedures involving animals in this study were approved by the Thammasat University Animal Care and Use Committee in Pathum Thani, Thailand, under Animal Ethics number 003/2020. The detail of the treated animals, including the experimental groups and treatments, were described by Lapmanee et al. (2023a). Briefly, forty-five adult male Wistar rats (Nomura Siam International Company Limited, Bangkok, Thailand) were housed in groups (2–3 rats/cage) to prevent individual stress and fear under 12 h light (245 ± 5 lux)/12 h dark cycle in an ambient temperature of 25 ± 2 °C and a humidity level of 55 ± 5%. Animals were fed *ad libitum* with standard diet (CP Company Limited, Bangkok, Thailand) and had unrestricted access to water. After 7 days of acclimatization, rats were randomly divided into five groups, each consisting of an equal number of animals (9 rats/group): (i) healthy control group (Con), (ii) stressed group (Str), (iii) stressed+Vlx group (Str+Vlx), (iv) stressed+Syn group (Str+Syn) and (v) stressed+Vlx+Syn group (Str+Vlx+Syn). Stressed rats were induced to exhibit depressive- and memory impairment-like behaviors by being confined in plastic cones to restrict their movement for 2 h each day at 9:00 AM and 11:00 AM and then returned to their home cage for 14 days. Based on the previous studies by Lapmanee et al. (2017, 2023a), this study freshly prepared all the regimens and utilized sterile normal saline to dissolve synbiotics and/or antidepressants; therefore, normal saline served as the vehicle. Con and Str groups were subjected to oral gavage with normal saline vehicle. In Vlx-treated animals, the antidepressant drugs Vlx (Pfizer Ireland Pharmaceuticals, Co. Kildare, Ireland) a dose of 10 mg/kg body weight was administered orally once daily. In Syn-treated animals, Syn (Inter Pharma Public Co., Ltd., Bangkok, Thailand) containing $1.0 \times 10^{10}$ CFU of probiotic strains (including *Bifidobacterium bifidum, Bifidobacterium infantis, Bifidobacterium longum, Lactobacillus acidophilus, Lactobacillus casei*, and *Lactococcus lactis*) and oligosaccharide in a 2-gram sample as determined by series dilution, was administered orally once a day. Both Vlx and Syn were freshly prepared and administered daily at 3:00 to 4:00 PM for 14 days. If animals displayed signs of illness, lack of food or water intake, excessive body weight loss exceeding 20%, or poor ambulation in their home cage, rats were excluded from the experiments. Fortunately, no such incidents occurred in the present study. Consequently, the final number of rats per group at the conclusion of the study remained consistent at nine rats.

The body weight of all animals was monitored daily, and they were individually housed in metabolic cages for 24 h on a weekly basis for the assessment of food and water intake. Following the induction of stress and the administration of Vlx and Syn, all rats underwent a series of behavioral tests, including the MWM on days 15 to 17 and FST on day 18. Blind conditions of treatment between analyzers were implemented in the behavioral test.

In each group, four rats underwent oral gavage with fluorescein isothiocyanate (FITC) dextran 1 h before blood collection to study intestinal permeability. The rats were ultimately euthanized using an overdose of isoflurane anesthesia, and subsequently, blood and specific target organs, *i.e.*, the brain and ileum, were harvested for analysis. Whole brains were then rapidly extracted, frozen in liquid nitrogen, and stored at −80 °C for GDNF protein studies. For histomorphological analyses, rats underwent transcardiac perfusion with PBS (pH 7.4) and 4% paraformaldehyde (PFA), followed by 24-h PFA fixation. The processed tissues were embedded in paraffin for further studies. Additionally, urine was collected from the bladder to measure corticosterone (CORT) levels. Wet weights of the adrenal glands, spleen, and thymus were harvested and assessed as markers of stress and immune responses, as previously described (*Resendez & Rehagen, 2017*; *Lapmanee et al., 2017*, *2023a*). Changes in blood glucose, total cholesterol, triglycerides, and total fat pad weight were recorded as indicators of physical changes and metabolic health (*Hyvärinen et al., 2022*). The experimental design is shown in Fig. 1.

## Morris water maze test

The MWM was utilized to assess learning and memory performance, following the method outlined by *Lapmanee et al. (2017)*. The MWM comprises a circular stainless-steel pool, 150 cm in diameter and 60 cm in height, filled with tap water at a controlled temperature of 23 ± 2 °C, rendered opaque by the addition of 250 mL of milk. Surrounding the pool are cues for spatial orientation, such as black/white circles, crosses, and a black grid. Inside the pool, there is a stainless-steel platform, measuring 10 cm in diameter and positioned 30 cm below the water's surface, serving as the target. In each trial, a rat was gently placed into one of the four quadrants of the maze: North, South, East, or West. Prior to the commencement of the learning trials, each rat was briefly allowed to become acquainted with the platform by being placed on it for 10 s. If a rat moved away from the platform before the 10 s had elapsed, it was gently returned to its home cage. (i) Spatial learning was assessed through measurements of escape latency. The platform's location remained constant. If a rat could not locate the platform within 60 s, it was gently placed on the platform, and an escape latency of 60 s was recorded. This training extended over three days, during which the rats completed a total of 20 trials: eight trials on the first two days and four trials on the third day, with 5-min breaks between each trial. An increase in escape latency indicated poorer spatial learning. (ii) The probe test occurred on the third day after training sessions, each lasting 1 h. During this test, the platform was removed from the pool. Spending more time in the correct quadrant (where the platform had been located) indicated better spatial memory.

## Forced swimming test

Each rat underwent two swimming sessions within 24 h in a cylindrical container (45 cm height, 25 cm diameter) filled with tap water at 25 ± 2 °C to a depth of 35 cm. During the first 15-min session, the rat was assessed for immobility, swimming, and climbing behaviors. The immobility time in the FST, characterized by the minimal movement to keep the head above water while floating, reflects behavioral despair—a symptom of

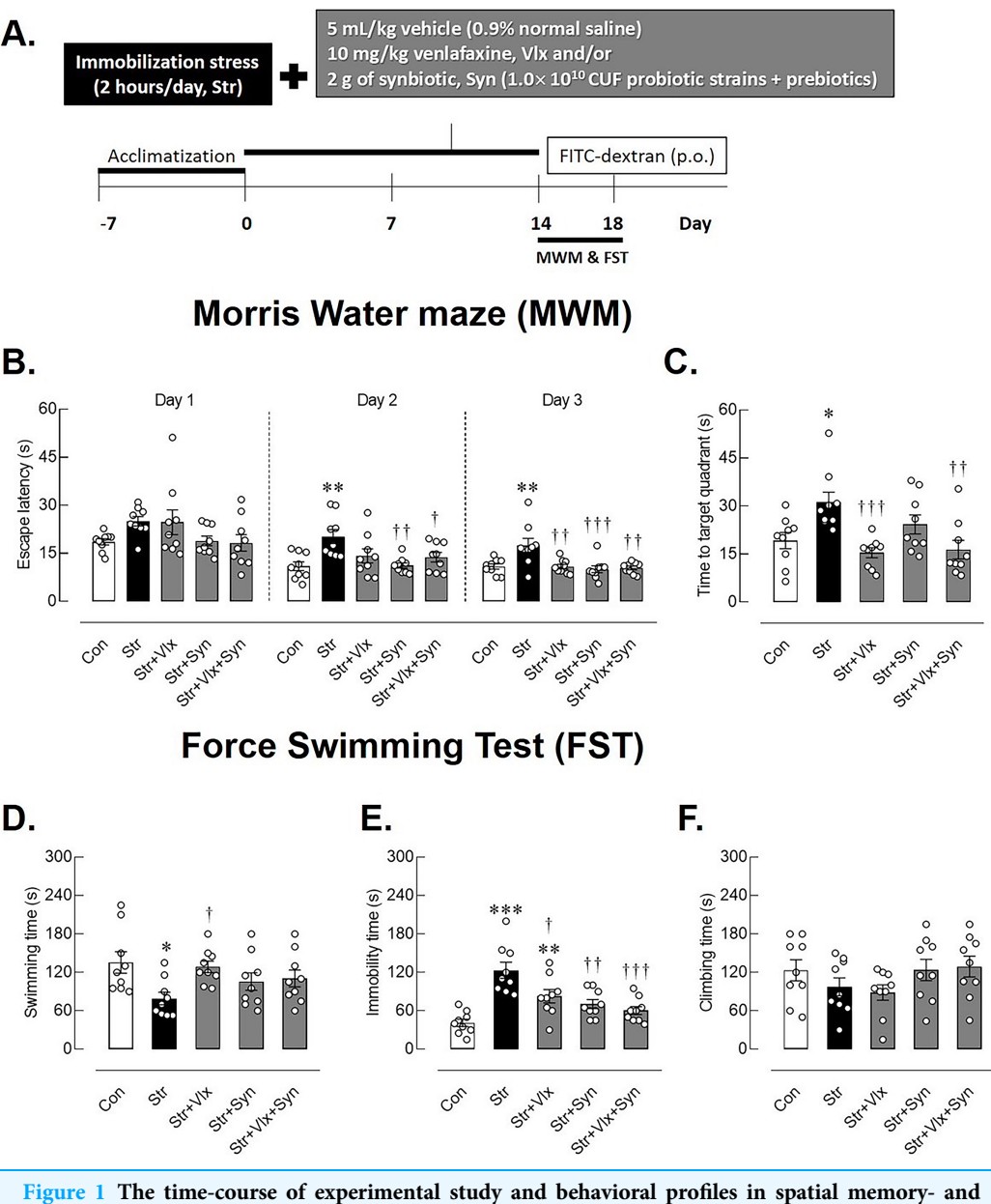

**Figure 1 The time-course of experimental study and behavioral profiles in spatial memory- and depression-like behaviors as determined by Morris Water Maze (MWM) and forced swimming test (FST).** Experimental procedures (A), escape latency in learning trails (B), correct quadrant time in probe test (C), swimming time (D), immobility time (E), climbing time (F) in male rats under stress exposures. $^*p < 0.05$, $^{**}p < 0.01$ and $^{***}p < 0.001$ compared to vehicle-treated control rats. $\dagger p < 0.05$, $\dagger\dagger p < 0.01$ and $\dagger\dagger\dagger p < 0.001$ compared to vehicle-treated stressed rats ($n = 9$ rats/group Con: control; Str: stressed+vehicle; Str+Vlx: stressed+venlafaxine; Str+Syn: stressed+synbiotic supplement; Str+Vlx+Syn: stressed+venlafaxine+synbiotic supplement.

depression. Swimming involved active forepaw movement, including crossing between quadrants and turning. Climbing was defined as upward forepaw movements along the cylinder's inner wall. An increase in the duration of immobility or a decrease in the duration of swimming or climbing behaviors were indicative of depression-like behaviors in the rats (*Lapmanee et al., 2017*; *Songphaeng et al., 2023*).
## Body weight gain, food and water intake

Body weights were measured daily. To calculate the daily body weight gain for each rat, the difference between the final body weight (Day 14) and the initial body weight (Day 1) was divided by 14 days. For food and water intake, measurements were recorded weekly during the experiment. Measurements were taken by weighing the food and measuring the water volume before and after a 24-h period in the metabolic cage (model 3700M071-01CS; Tecniplast, Varese, Italy).

## Euthanasia

The rats were euthanized through an overdose of anesthesia inhalation. Isoflurane was used for anesthesia in spontaneously breathing rats. Each rat was situated in a plastic container with a top inlet, and a silicon tube providing a mixture of anesthetic and oxygen gas was connected. Anesthesia was induced by inhaling 5% vaporized isoflurane. It is crucial to note that there were no noticeable movements or reactions to pain stimuli observed in the animals before the commencement of specimen collection.

## Blood and urinary biochemical assays

Blood was collected *via* cardiac puncture between 9:00 AM and 12:00 PM, and serum and plasma were subsequently prepared. Serum glucose, total cholesterol, and triglycerides were assayed using standard methods on an automated clinical chemistry analyzer (Fuji Dri-CHEM NX500i, Tokyo, Japan). Plasma was analyzed for FITC-dextran assay. Rat urinary CORT levels were determined through commercial radioimmunoassay provided by Immunodiagnostic Systems Ltd. (Tyne and Wear, UK).

## FITC-dextran assay

After 24 h from the last behavioral test, rats were orally administered 70-kDa FITC dextran (Sigma-Aldrich, St. Louis, MO, USA) at a dose of 25 mg/kg, as reported by *Xia et al. (2021)*. Subsequently, 1 h later, blood was collected into anticoagulant tubes treated with EDTA for plasma preparation. Then, blood samples were vertically kept at room temperature for 15 min. For plasma isolation, the upper parts of blood samples were centrifuged at 2,000 × g for 10 min. Plasma was diluted at a 1:5 (v/v) ratio in phosphate-buffered saline (pH 7.4). Fluorescence intensity was spectrophotometrically measured in 96-well plates using an excitation wavelength of 485 nm and an emission wavelength of 528 nm.

## Tissues and organ collections

After completing the behavioral tests, animals were anesthetized using an overdose of isoflurane inhalation. Brains and ileum were carefully dissected free of surrounding tissues, and this dissection process was conducted by a single investigator to maintain consistency. Additionally, the adrenal glands, spleen, thymus, total fat pads from the abdominal cavity, and the area representing fat accumulation, including visceral, retroperitoneal, and epididymal regions, were individually dissected and weighed relative to the final body weight using a similar method. Furthermore, the adiposity index was calculated as the sum of total fats divided by the final body weight, multiplied by 100 (*Leopoldo et al., 2016*).

## Hippocampal and ileum histomorphology

Histological examination of sections stained with hematoxylin and eosin (Sigma-Aldrich, St. Louis, MO, USA) revealed characteristic features in the hippocampal dentate gyrus and small intestine ileum. Tissues from the brain and ileum of four rats in each group were harvested and then preserved in 4% PFA. The target tissues were dehydrated using serial ethanol, cleaned in xylene, and embedded in paraffin. Coronal sections, each with a thickness of 5 μm, were then cut using a microtome (Microsystems, Wetzlar, Germany), mounted on glass slides, and stained using the routine hematoxylin and eosin protocol. Cell death was assessed by the presence of pyknotic cells in the dentate gyrus, while inflammatory cell infiltrate was scored in the ileum (*Erben et al., 2014*; *Akang, 2019*). Histological slide interpretations were performed through a blind analysis by two pathobiologists. Each observed intestinal injury was assigned a score, and a cumulative score was calculated at the end of each investigation. The scoring system ranged from 0 to 25 points, with zero indicating no signs of injury to the animals and 25 representing maximum injury, such as the loss of villi. Additionally, the number of Peyer's patches, crucial to mucosal immunity and modulation of intestinal immune and inflammatory responses, was also observed (*Jung, Hugot & Barreau, 2010*). Examination of slides, photography, and morphometric studies were conducted at the Pathology Information and Learning Center, Department of Pathobiology, Faculty of Science, Mahidol University, Thailand.

## Western blot analysis

The brain was carefully extracted from the skull and the ileum was cleaned with sterile PBS. The dissection of the hippocampus followed the protocols outlined by *Lapmanee et al. (2023b)*. Subsequently, the hippocampal and ileum tissues were combined with a lysis buffer containing protease and phosphatase inhibitor cocktails (Sigma, Burlington, MA, USA). The total protein concentration was quantified using the BCA Protein Assay kit (Thermo Scientific Inc., Waltham, MA, USA). Then, the protein samples (50 μg each) were loaded into the wells of 4–20% Mini-PROTEAN TGX Precast Gels or 15% SDS-PAGE gels depending on their molecular weights and subsequently transferred onto PVDF membranes pre-soaked in methanol (Amersham Biosciences, New Jersey, USA). These membranes were then subjected to overnight incubation at 4 °C with either a 1:1,000 dilution of rabbit polyclonal anti-GDNF antibody (catalog no. Cat #PA5-89957; Invitrogen, Thermo Fisher Scientific, Waltham, MA, USA), a 1:5,000 dilution of rabbit polyclonal anti-β-actin antibody (catalog no. AB8227; Abcam, Cambridge, UK) or GAPDH antibody (catalog no. Cat #PA1-988; Invitrogen, Thermo Fisher Scientific, Waltham, MA, USA). Following thorough washing steps with TBST, the membranes were exposed to a 1:5,000 dilution of goat anti-rabbit IgG H&L (HRP) secondary antibody (catalog no. AB205718; Abcam, Cambridge, UK) at room temperature for 2 h. Protein bands were visualized using an ECL Prime Western blot detection reagent with the image reader Amersham ImageQuant 500. Densitometry analysis was conducted by software (CYTIVA, Marlborough, MA, USA). The GDNF protein expression level in relation to β-actin or GAPDH in the control group was subsequently normalized to a value of 1.

## Statistical analysis

The data are reported as means ± standard error of the mean (SEM). The two data sets were compared using an unpaired Student's t-test. Group differences were evaluated using one-way analysis of variance (ANOVA) with Dunnett's multiple comparison test. Statistical significance was set at $p < 0.05$. All statistical analyses and graphical representations were conducted using GraphPad Prism 8 (GraphPad Software Inc., San Diego, CA, USA).

# RESULTS

## Syn and Vlx supplementation improves physiological and biochemical parameters changes in stressed rats

As summarized in Table 1, immobilization stress significantly reduced body weight gain ($p < 0.01$) and daily food intake ($p < 0.001$) in the stressed rats when compared to non-stressed rats. In contrast, stressed rats exhibited increased water intake ($p < 0.001$), likely to maintain hydration and eliminate waste. The stressed rats had significantly elevated urinary CORT levels ($p < 0.0001$) and an increased ratio of wet adrenal gland weight to body weight ($p < 0.05$) when compared to the control group. Although blood glucose and total cholesterol remained unaffected, stressed rats had higher triglyceride levels ($p < 0.05$) as compared to the control group, indicating fat breakdown due to stress hormones and decreased adiposity index ($p < 0.05$). Additionally, immobilized stress reduced thymus weight but not spleen weight ($p < 0.05$), potentially impairing immune function. Thus, immobilization stress triggers complex physiological responses affecting stress hormones, metabolism, appetite regulation, and immune function, underscoring the multi-system impact of stress in animals.

Supplementation with either Syn or Vlx attenuated stress-induced body weight loss and restored food and fluid intake to normal levels in non-stressed rats. Syn treatment significantly reduced urinary CORT levels ($p < 0.0001$) in stressed rats, similar to the effect of Vlx treatment. This results show that both Syn and Vlx modulates of neurotransmitter activity in body weight regulation during stress. Both Syn alone or the combination of Syn and Vlx treatments significant reduced blood glucose ($p < 0.001$) and total cholesterol ($p < 0.05$) in stressed rats, indicating potential improvement in lipid metabolism due to synbiotic supplementation.

## Syn and Vlx supplementation attenuates the stress-induce depression- and memory impairment-like behaviors

MWM was performed to evaluate animals' learning behavior and spatial memory after 14 days of stress by immobilization and treatments. During the training phase (Fig. 1B), stressed rats exhibited extended escape latency compared to the non-stressed rats, particularly on days 2 ($p < 0.01$) and day 3 ($p < 0.001$). In contrast, stressed rats treated with Vlx and/or Syn supplements displayed reduced time to reach the platform on days 2 ($p < 0.01$). All treatments also reduced time spent swimming to the platform on day 3 ($p < 0.001$). Additionally, Fig. 1C shows that Vlx and the combination of Syn and Vlx treatment increased the time spent swimming toward the target quadrant ($p < 0.001$),

**Table 1  Alteration of food and water consumption, body, organ weights and biochemical profiles in male rats under stress exposures.**

| | Con | Str | Str+Vlx | Str+Syn | Str+Vlx+Syn |
|---|---|---|---|---|---|
| Body weight gain (g) | 8.99 ± 0.11 | 7.84 ± 0.18** | 8.73 ± 0.26† | 8.76 ± 0.25† | 8.55 ± 0.20 |
| Daily food intake (g) | 28.11 ± 0.63 | 24.11 ± 0.54*** | 26.44 ± 0.77† | 27.56 ± 0.44††† | 26.44 ± 0.60††† |
| Daily water intake (mL) | 14.44 ± 0.41 | 17.44 ± 0.38*** | 15.00 ± 0.41†† | 15.44 ± 0.50† | 15.67 ± 0.53† |
| Adiposity index | 10.57 ± 0.46 | 7.88 ± 0.72* | 9.08 ± 0.55 | 9.04 ± 0.48 | 8.73 ± 0.44 |
| % Adrenal weight | 0.02 ± 0.00 | 0.03 ± 0.00** | 0.02 ± 0.00† | 0.03 ± 0.00 | 0.02 ± 0.00 |
| % Spleen weight | 0.21 ± 0.01 | 0.19 ± 0.01 | 0.20 ± 0.01 | 0.19 ± 0.01 | 0.20 ± 0.02 |
| % Thymus weigh | 0.20 ± 0.01 | 0.16 ± 0.01* | 0.19 ± 0.01 | 0.17 ± 0.01 | 0.18 ± 0.01 |
| Glucose (mg/dL) | 120.40 ± 7.95 | 143.30 ± 8.84 | 119.80 ± 7.45 | 100.70 ± 4.32††† | 104.7 ± 6.86†† |
| Cholesterol (mg/dL) | 72.44 ± 5.00 | 82.00 ± 7.14 | 77.33 ± 9.68 | 60.33 ± 3.18† | 61.58 ± 2.56† |
| Triglycerides (mg/dL) | 108.10 ± 4.18 | 157.20 ± 8.13* | 118.80 ± 17.97 | 137.90 ± 7.17 | 132.00 ± 13.58 |
| Urinary CORT (ng/mL) | 14.22 ± 1.51 | 30.63 ± 2.05*** | 15.42 ± 1.09††† | 21.49 ± 0.89**††† | 15.28 ± 1.21††† |

Notes:
Data are expressed as mean ± SEM. Organ weights are presented as a percentage of wet mass relative to the final body weight.
* $p < 0.05$.
** $p < 0.01$.
*** $p < 0.001$ compared to vehicle-treated control rats.
† $p < 0.05$.
†† $p < 0.01$.
††† $p < 0.001$ compared to vehicle-treated stressed rats ($n$ = 9 rats/group). CORT, corticosterone; Con, control; Str, stressed+vehicle; Str+Vlx, stressed+venlafaxine; Str+Syn, stressed+synbiotic supplement; Str+Vlx+Syn, stressed+venlafaxine+synbiotic supplement.

suggesting the potential of Syn and Vlx to improve learning and spatial memory in stressed rats.

In Fig. 1D, stressed rats demonstrated significantly shorter swimming times compared to the control group ($p < 0.05$), indicating the manifestation of hopelessness behavior in the FST, a recognized symptom of depression. Interestingly, even though stressed rats treated with Vlx exhibited more immobility time than the control rats ($p < 0.0001$), they still displayed reduced depression symptoms, as demonstrated by an increase in swimming time (Fig. 1E). Both Syn and Vlx supplementation effectively mitigated depression-like behaviors, as indicated by the reduced immobility time in stressed rats ($p < 0.001$). Importantly, there were no differences in climbing behavior among the groups, suggesting that stress induction and subsequent treatments did not significantly impact motor activity (Fig. 1F). These results suggest that Syn and Vlx supplementation ameliorate stress-induced depression-like behaviors without altering motor activity.

## Syn and Vlx supplementation ameliorates stress-induced neuronal cell death in rat hippocampus

Stress induced markedly changes in the histomorphology of the hippocampus, specifically in the dentate gyrus of male rats, contrasting with the control group. A normal granular cell, characterized by a dark-staining nucleus and scanty cytoplasm, is denoted by the black arrow. Pyknotic nuclei, indicating small, condensed, dark-staining nuclei with eosinophilic cytoplasm, were more prevalent in stressed rats, demonstrating a significantly higher percentage of pyknotic cells in the dentate gyrus ($p < 0.0001$), indicative of increased neural cell death (see Fig. 2F).

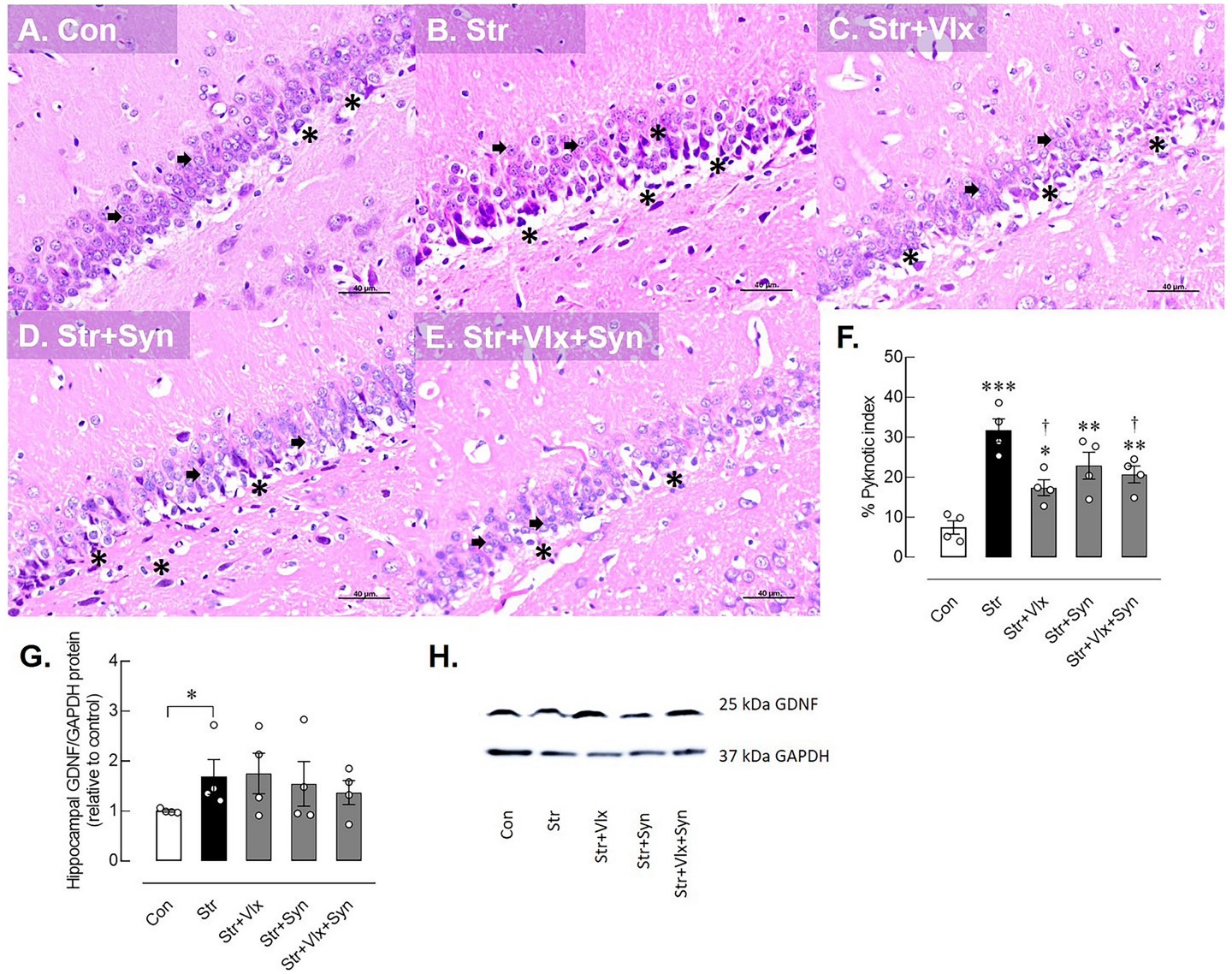

**Figure 2 Histomorphological and neuroprotective protein profiles of hippocampus as determine by H&E stating and Western blotting analysis.** Histological images of hippocampus (A–E) and pyknotic index in the dentate gyrus of the hippocampus in male rats under stress exposures (F). The black arrow indicates a normal granular cell, characterized by a dark-staining nucleus and scanty cytoplasm. The black asterisk indicates pyknotic nuclei, small condensed, dark-staining nuclei with eosinophilic cytoplasm. Relative glial cell-derived neurotrophic factor (GDNF) protein in hippocampus in stressed rats (G) and the representative expression of GDNF to GAPDH (H). $^{*}p < 0.05$, $^{**}p < 0.01$ and $^{***}p < 0.001$ compared to vehicle-treated control rats. $†p < 0.05$ compared to vehicle-treated stressed rats ($n = 4$ rats/group). Con, control; Str, stressed+vehicle; Str+Vlx, stressed+venlafaxine; Str+Syn, stressed+synbiotic supplement; Str+Vlx+Syn, stressed+venlafaxine+synbiotic supplement.

As anticipated, Vlx exhibited neurotherapeutic effects against stress-induced neuronal cell death compared to vehicle treatment. Furthermore, the combination of Vlx and Syn attenuated nerve cell death, reducing the pyknotic index percentage compared with stressed vehicle-treated rats ($p < 0.05$).

## Syn and Vlx supplementation improves intestinal permeability and inflammation in the gut of stressed rats

The morphology of the ileum revealed desquamation of the villi and villous edema in the control group. In contrast, the stressed group exhibited a pronounced loss of villi, along with villous edema and extensive layer separation. Inflammatory signs, including the presence of Peyer's patches, were observed in the vehicle-treated rats, but these signs were not observed in the other treatment groups (Supplemental Figure). As a result, restraint stress significantly elevated the inflammation score in ileum histomorphology compared to the control group ($p < 0.0001$). Furthermore, variations in villi numbers, desquamation, flattened villi, and edema layer were observed across all interventions. Despite these commonalities, Syn displayed a mitigated ileum inflammatory score compared to the vehicle-treated group ($p < 0.05$, as shown in Fig. 3F).

The results of the intestinal permeability study (Fig. 3G) indicated that stressed vehicle-treated rats had higher plasma FITC-dextran levels than the control group ($p < 0.0001$), indicating a leaky gut condition in stressed rats. Significantly, both Vlx and Syn supplementation were more effective in reducing FITC-dextran levels in stressed rats than the vehicle treatment ($p < 0.0001$). Furthermore, Syn supplementation appears to reduce intestinal inflammation in stressed rats. Vlx may ameliorate stress-induced intestinal hyperpermeability through modulation of 5-HT and NE activity.

## Syn and Vlx supplementation enhances the effect of stress on GDNF expression levels in the hippocampus and gut of rats

The possible molecular mechanisms underlying cellular changes in the hippocampus and ileum of stressed rats were examined (Fig. 2). The expression of the GDNF protein significantly increased in stressed rats ($p < 0.0495$). In Figs. 2G–2H, treatment with Vlx and/or Syn did not alter hippocampal GDNF protein levels compared to controls or the vehicle treatment group. In the ileum (Fig. 3), stress slightly upregulated GDNF protein levels ($p = 0.0525$), whereas all treatments were trending to restore GDNF protein levels to controls (Figs. 3H and 3I). These findings suggest that stress negatively impacts brain and intestinal health, and both Vlx and Syn may protect against neural cell death in the dentate gyrus of the hippocampus. Additionally, Syn could offer partial protection by modulating GDNF expression, contributing to the restoration of the disrupted gut barrier caused by stress exposure in male rats.

## DISCUSSION

Imbalances in gut microbiota among individuals experiencing stress and psychiatric patients can lead to dysfunction in the gut-brain axis. Exposure to stressors contributes to the long-term hyperactivity of the HPA axis, resulting in the prolonged release of glucocorticoids (*Fowler, Bogdan & Gaffrey, 2021*). These stress hormones have a profound impact on a wide range of physiological and behavioral dimensions (*Francois et al., 2022*; *Cheon, 2023*).

In the present study, stressed rats exhibited reduced body weight gain and food intake, along with increased fluid intake, urinary corticosterone levels, and adrenal weight. These

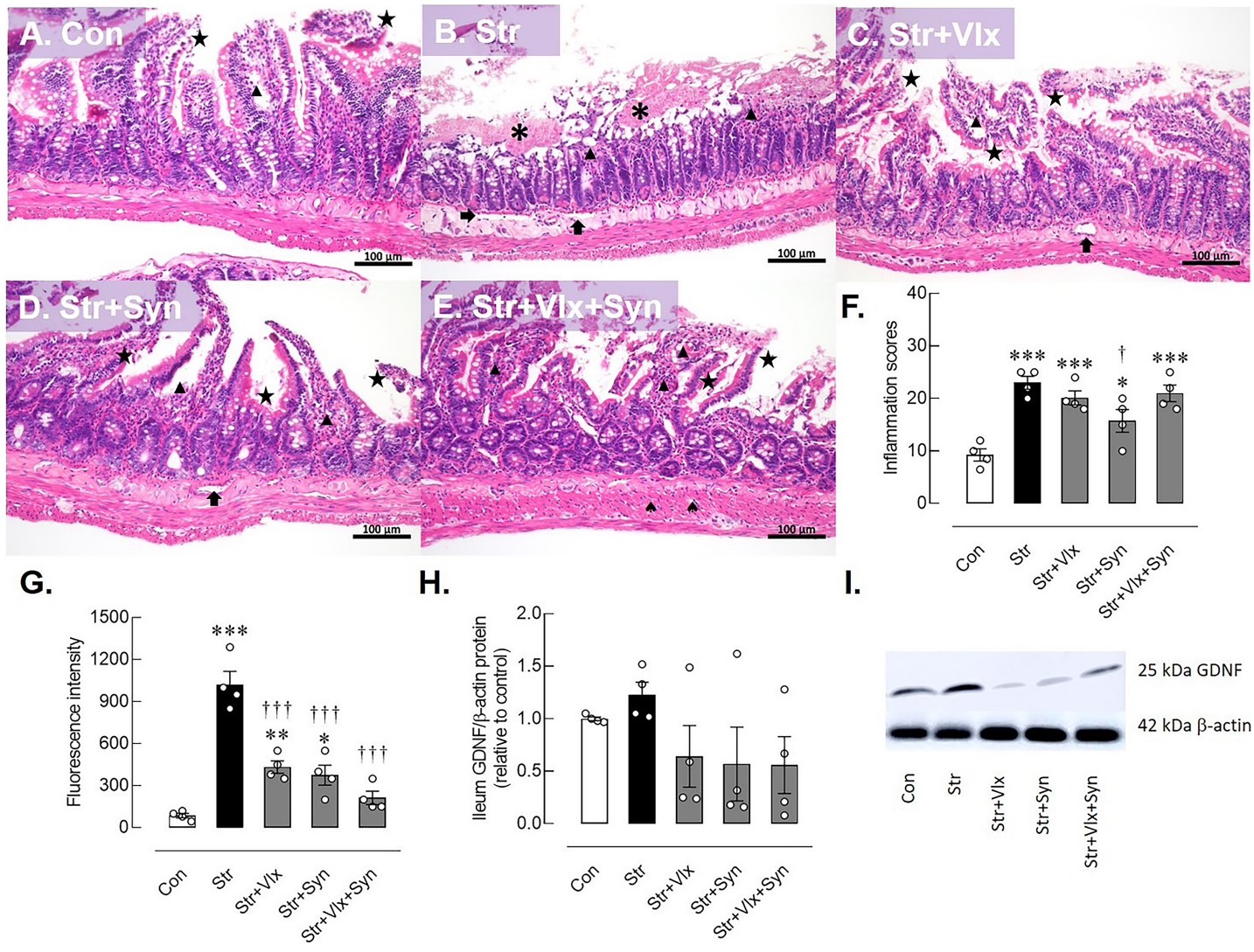

**Figure 3 Histomorphological, intestinal barrier-related function and neuroprotective protein profiles of ileum as determine by H&E stating, fluorescence assay and Western blotting analysis.** Histological images of ileum (A–E) and inflammatory scores in male rats under stress exposures (F). The morphology of the ileum revealed desquamation of the villi (black star), villous edema (black triangle), loss of villi (asterisks) and layer separation (black arrow). 70-kDa FITC-dextran intestinal permeability (G), relative glial cell-derived neurotrophic factor (GDNF) protein in ileum in stressed rats (H) and the representative expression of GDNF beta actin (I). *$p < 0.05$, **$p < 0.01$ and ***$p < 0.001$ compared to vehicle-treated control rats. †$p < 0.05$ and †††$p < 0.001$ compared to vehicle-treated stressed rats ($n = 4$ rats/group). Con, control; Str, stressed+vehicle; Str+Vlx, stressed+venlafaxine; Str+Syn, stressed+synbiotic supplement; Str+Vlx+Syn, stressed+venlafaxine+synbiotic supplement.

findings align with previous studies that have characterized reductions in body weight and appetite as common symptoms in stressed rodents and clinical human subjects (*Jeong, Lee & Kang, 2013*; *Lapmanee et al., 2017*). The relationship between stress hormones and energy metabolism is complex, and the results in this regard are often controversial and inconclusive. Stressed rats also showed a decrease in adiposity index, indicating increased fat metabolism, which suggests elevated levels of cholesterol, triglycerides, hepatic and serum inflammatory factors, and adipocytokines. This implies that stress hormones induce lipolysis and facilitate fat metabolism (*Liu et al., 2014*).

Furthermore, the induction of stress in the study may indirectly impact immune function, as evidenced by the reduced thymus weight and pro-inflammatory cytokines (*Gruver & Sempowski, 2008*; *Boersma et al., 2014*; *Seiler, Fagundes & Christian, 2020*). Reduced immune function during stressful conditions can disrupt the bidirectional communication between gut microbes and central monoamine neurotransmission, potentially contributing to the development of mood disorders and cognitive impairments (*Foster, Rinaman & Cryan, 2017*; *Appleton, 2018*; *Socała et al., 2021*). Elevated levels of glucocorticoids and compromised immune responses in animal stress models and patients are associated with anxiety and depression (*Frank, Watkins & Maier, 2013*; *Correia, Cardoso & Vale, 2021*). In individuals with depression, specific bacterial strains, *i.e.*, *Burkholderia oklahomensis*, *Ralstonia spp.*, *Acinetobacter baumannii*, and *Klebsiella spp.*, have been linked to the activation of metabolic pathways involved in 5-HT and nitric oxide synthesis (as reviewed by *Donoso et al. (2023)*). These physiological changes are reflected in the alterations in depression and spatial memory observed in stressed male rats.

Moreover, intestinal hyperpermeability and systemic inflammation are common in individuals exposed to stress (*Ilchmann-Diounou & Menard, 2020*). This is mainly due to glucocorticoids directly increasing inflammatory signals, affecting neurons, intestinal cells, and tight junction proteins (*e.g.*, claudin and occluding). Consequently, these stress-induced changes reduce the levels of claudin-1, occludin, and ZO-1 proteins, leading to greater permeability in the ileum and colon epithelium (*Mazzon et al., 2002*; *Zheng et al., 2013*). Interestingly, subacute stress in the jejunum has been linked to increased expression of claudin 1, 5, and 8, occludin, and ZO-1 mRNAs (*Lauffer et al., 2016*). All these interactions ultimately compromise the intestinal barrier, resulting in heightened permeability and inflammation (*Craig et al., 2022*; *Yu et al., 2022*).

Therefore, the stress procedures employed in the present study, involving 2 h of daily restraint stress for 14 days, have proven successful in eliciting physiological, behavioral, and biochemical responses in male rats (*Lee, 2015*; *Lapmanee et al., 2023a*; *Songphaeng et al., 2023*). Regarding behavioral responses, stressed male rats exhibited episodes of major depression, which were characterized by manifestations of depressive behavior and heightened emotionality. In addition to the FST, this study conducted a weekly sucrose consumption test to assess anhedonic-like behavior which is considered a sign of depression (*Songphaeng et al., 2023*). The results revealed that male rats exposed to stress exhibited heightened emotionality, as indicated by their higher sucrose consumption, with a tendency to decrease in the second week of the study. However, there was no significant difference in sucrose consumption among the treatment groups (see Supplemental Figure). These findings underscore the intricate relationship between stress, the gut-brain axis, and immune function, influencing various aspects of physiological and behavioral health. Understanding these mechanisms holds the potential to inform the development of interventions and treatments for conditions such as anxiety, depression, and metabolic disorders, which are influenced by stress-induced changes in both the gut and the brain.

While various monoaminergic antidepressants are frequently prescribed for individuals experiencing stress and psychiatric disorders, long-term effectiveness is limited, and they may be accompanied by specific side effects (*Santarsieri & Schwartz, 2015*). However, the

understanding of the interactions between antidepressants and the microbiota-gut-brain axis remains limited. Based on prior study, SNRIs such as duloxetine, milnacipran, and Vlx have demonstrated the ability to simultaneously increase the levels of neurotransmitters 5-HT and NE. Vlx has exhibited superior beneficial effects in alleviating anxiety-like behaviors, learned fear, stress-related behaviors, and promoting the expression of BDNF in stressed male rats (*Lapmanee et al., 2017*, *2023a*). These findings suggest that SNRIs, especially Vlx, may provide a more promising treatment option for central monoaminergic neurotransmission and neurotropic factors compared to other alternatives. However, the use of Vlx had an impact on the microflora and essential microbes in the intestine (*Shen et al., 2023*).

As shown in the results, Vlx restored corticosterone levels, partially mitigated tight junction protein dysfunction, and exhibited the potential to alleviate neuronal cell death in the dentate gyrus. Furthermore, Vlx was observed to induce the activation of EPO/EPOR/JAK2 signaling pathways, which contribute to its anti-apoptotic, anti-inflammatory, and neurotrophic effects (*Lapmanee et al., 2017*; *Saad et al., 2019*). Therefore, Vlx is effective in reducing behavioral abnormalities and restoring intestinal permeability.

In addition, Syn is a combination of prebiotics and probiotics, believed to exert a synergistic effect by inhibiting the growth of pathogenic bacteria and promoting the growth of beneficial organisms. This combination consists of mixed oligosaccharides and bacterial colonies comprising *Bifidobacterium bifidum*, *Bifidobacterium infantis*, *Bifidobacterium longum*, *Lactobacillus acidophilus*, *Lactobacillus casei*, and *Lactococcus lactis*, all of which were utilized in the present study. Remarkably, *Bifidobacterium bifidum* and *Lactobacillus acidophilus* strains have demonstrated the potential to mitigate the risk of antibiotic-induced diarrhea and alleviate symptoms of anxiety-like behaviors and novel object recognition impairment (*Kopacz & Phadtare, 2022*; *Lapmanee et al., 2023a*). Therefore, there is a need to investigate the impact of Syn in enhancing gut microbiota and explore the co-treatment of Vlx and Syn for central and intestinal protection, with the aim of reducing stress-induced depression and memory impairment.

On the other hand, Syn supplementation displayed promising effects in mitigating depression-like behaviors and spatial memory impairment, along with the reduction of corticosterone levels and improvement in intestinal permeability and inflammation. In terms of behavior control and memory formation, Syn upregulated 5-HT1A mRNA levels in the dentate gyrus and cornu ammonis one layer, indicating its potential to protect against neuronal cell death (*Barrera-Bugueño et al., 2017*). Additionally, Syns have the capacity to modulate the gut microbiota ecosystem, and these mechanisms could increase the levels of monoamines and their metabolites while attenuating hyperpermeability and inflammation markers (*i.e.*, TNF-$\alpha$, IL-6, IL-18, and IL-1$\beta$) in stressed rats (*Huang et al., 2022*). Nevertheless, the impact of this relationship on compromising the integrity of the intestinal barrier in stress-related neuropsychiatric conditions, which can lead to dysregulation in the endocrine, neural, and immune systems, remains a relatively unexplored area of research.

In addition to assessing ileum inflammation and Peyer's Patches, the villi-to-crypt ratio of the duodenum was determined (Supplemental Figure). Stressed rats displayed a trend

towards an increased ratio, potentially indicating compensatory responses to stress. In the ileum, Syn reduced inflammation and restored the function of the epithelial barrier, which is linked with the gut microbiota (*Kelly et al., 2015*). Nevertheless, the probiotics in Syn, such as *Lactobacillus reuteri*, have the potential to improve tight junction protein expression. This improvement is associated with the compromise of the integrity of the intestinal barrier in stress-related neuropsychiatric conditions, which can lead to dysregulation in the endocrine, neural, and immune systems (*Rose et al., 2021*).

A potential mechanism involves the regulation of neurotrophic factors such as GDNF, NGF, and BDNF. GDNF is particularly crucial for promoting neuronal survival and protecting against stress-related damage (*Shen et al., 2019*). While GDNF plays a role in neuroprotection and neuroplasticity in the central and peripheral nervous system, other factors such as BDNF, ciliary neurotrophic factor, NGF, or neurotrophin-3 can directly impact protective barrier–environment interactions along the gut–brain axis, influencing cognition and behaviors related to anxiety and depression during stress exposure (*Liu, 2018*; *Lapmanee et al., 2017*, *2023a*). The upregulation of GDNF expression in stressed rats suggests a highly protective response to stress induction and related disorders (*Duarte Azevedo, Sander & Tenenbaum, 2020*). While neither Vlx nor Syn showed clear and significant GDNF protein expression in the hippocampus or ileum of stressed male rats, there is another possibility to restore central and intestinal homeostasis and function. Interestingly, Vlx appears to enhance hippocampal GDNF expression while reducing it in the ileum, possibly linked to its central neuroprotective effects. In contrast, Syn demonstrates an effective capability to restore GDNF expression in the ileum, indicating its potential role in mitigating intestinal inflammation and supporting intestinal health.

To our knowledge, the combined Syn with Vlx treatment could be suggested as an intervention in stressed individuals at risk of developing depression and memory impairment. These interventions might improve the microbiota–gut–brain axis by modulating enteric and brain neurotrophic factors. Collectively, these findings suggest that Vlx has strong central neuroprotective effects against stress induction, while Syn potentially alleviates intestinal inflammation. Both treatments attenuate hippocampal neuronal cell death and improve intestinal health in stressed male rats through potential mechanisms involving the regulation of neurotropic factors.

## CONCLUSIONS

Stress disrupts the gut-brain axis, causing physiological and behavioral changes. The present study successfully induced stress responses in male rats. Stress can increase levels of pro-inflammatory cytokines, which can negatively affect tight junction proteins and compromise the integrity of the blood-brain and gut barriers, affecting both the brain and gut microbiota. SNRIs such as Vlx show potential in reducing stress-related behaviors and modulating GDNF expression, but it is important to consider their impact on gut microbiota. Furthermore, Syn has modulated the gut microbiota and facilitated central and intestinal protection in mitigating stress-induced depression and memory impairment. Both treatments reduce behavioral abnormalities, restore intestinal permeability, and influence central and enteric neurotrophic factors such as BDNF, NGF and neurotrophin-3.

This suggests potential therapeutic avenues, although precise mechanisms require further investigation.

## ACKNOWLEDGEMENTS

We thank Dr. Saranya Siribal, Mr. Somkiat Sarachat, and Miss Chayuda Tangsripongkul for their excellent assistance in tissue collection and sample preparation.

### Funding

This study was supported by the Faculty of Medicine, Siam University, to Sarawut Lapmanee (No. 002/2565) and to Nattakan Treesaksrisakul (No. 003/2565). The Target Development Group grant (Cosmeceuticals, No. P1952244) from the National Science and Technology Development Agency (NSTDA, Thailand), awarded to Mattaka Khongkow and Katawut Namdee, is also supported. Additionally, the study is financially supported by King Mongkut's Institute of Technology Ladkrabang (No. 2566-02-16-001) to Chaiyos Sirithanakorn. There was no additional funding received for this study. The funders had no role in study design, data collection and analysis, decision to publish, or preparation of the manuscript.

### Grant Disclosures

The following grant information was disclosed by the authors:
Faculty of Medicine, Siam University: 002/2565 and 003/2565.
The Target Development Group grant: P1952244.
National Science and Technology Development Agency, Thailand.
King Mongkut's Institute of Technology Ladkrabang: 2566-02-16-001.

### Competing Interests

The authors declare that they have no competing interests.

### Author Contributions

- Sarawut Lapmanee conceived and designed the experiments, performed the experiments, analyzed the data, prepared figures and/or tables, authored or reviewed drafts of the article, and approved the final draft.
- Nattapon Supkamonseni performed the experiments, authored or reviewed drafts of the article, and approved the final draft.
- Sakkarin Bhubhanil performed the experiments, authored or reviewed drafts of the article, and approved the final draft.
- Nattakan Treesaksrisakul performed the experiments, authored or reviewed drafts of the article, and approved the final draft.
- Chaiyos Sirithanakorn performed the experiments, analyzed the data, prepared figures and/or tables, authored or reviewed drafts of the article, and approved the final draft.
- Mattaka Khongkow performed the experiments, authored or reviewed drafts of the article, and approved the final draft.

- Katawut Namdee performed the experiments, analyzed the data, prepared figures and/or tables, authored or reviewed drafts of the article, and approved the final draft.
- Piyaporn Surinlert performed the experiments, authored or reviewed drafts of the article, and approved the final draft.
- Chittipong Tipbunjong performed the experiments, analyzed the data, authored or reviewed drafts of the article, and approved the final draft.
- Prapimpun Wongchitrat conceived and designed the experiments, performed the experiments, analyzed the data, prepared figures and/or tables, authored or reviewed drafts of the article, and approved the final draft.

### Animal Ethics

The following information was supplied relating to ethical approvals (*i.e.*, approving body and any reference numbers):

Thammasat University Animal Care and Use Committee in Pathum Thani, Thailand, under Animal Ethics number 003/2020.

### Data Availability

The raw measurements are available in the Supplemental File.

### Supplemental Information

Supplemental information for this article can be found online at http://dx.doi.org/10.7717/peerj.17033#supplemental-information.

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
