# Peer review of "Stress-induced changes in cognitive function and intestinal barrier integrity can be ameliorated by venlafaxine and synbiotic supplementations"

_PeerJ, doi:10.7717/peerj.17033_

## Round 0.1 · original submission · Minor Revisions

I agree with the reviewers, the submission is in need of minor revisions.

·

Basic reporting

The language used is clear in the discussion, however, some terms may be best clarified for non-pathologists. Some descriptions of how depressive phenotypes are increased and decreased need to be stated somewhat clearer.
Literature references are sufficient, the intersection between GDNF and other diseases beyond stress response may be important to highlight as well. The blots lack clarity in the figures a higher resolution tiff image may be needed.

Experimental design

The experiment clarifies a specific question. Some vehicle controls would be appreciated to increase scientific rigor in regards to the bacterial treatments, or an element referencing known effects of vehicle.

Please clarify the method to determine a lack of mobility problems that would complicate FST also if any additional depressive tests were done. The standard for euthanasia is often two factor including a gas followed by spinal dislocation, if this done in the protocol as it is not clear. Was the sample of brain tissue collected post perfusion or drop fixation? Please clarify if the CFU listed is for each colony total or for each species of bacteria, if it is total please describe the method to determine the amount of each species. Please specify the pathologist which has done the evaluation. An evaluation of the ileum's peyer's patches in regard to the inflammatory aspect of the core hypothesis may be an interesting element to the study.

Validity of the findings

The conclusions that GDNF is relevant to this specific mechanism seems to be weakly supported by the presented figures in regard to a direct effect on the brain, if included in conclusions this may need to be clarified as a indirect effect and proposed mechanism. The relevance of the Syn therapy showing slightly improved effect in comparison to the Vlx therapy might be interesting to note and explore statistically.

Reviewer 2 ·

Basic reporting

The manuscript presents the influence of two isolated treatments (venlafaxine or symbiotic) or their combination on behavioral, biochemical, and histological parameters in rats subjected to a stress protocol. The study aims to investigate whether these treatments improve stress-induced damage, taking into consideration the gut-brain axis connection.

Some considerations:

Please provide the specific reference for the first paragraph.

The English needs to be reviewed for clarity and coherence.

Experimental design

The experimental protocol was conducted to complement a study previously published by the research group. The tests performed are commonly used and well-described in the academic community working in this area.

Why wasn't periepididymal fat collected as an index of adiposity, given that it is more commonly described in the literature?

Please describe in the material and methods item, more clearly, how was collected this tissue (adipose tissue).

Validity of the findings

The data presented appears to be clear, well-described in the manuscript, and the figures are self-explanatory. All data is succinctly discussed in the corresponding section.

Reviewer 3 ·

Basic reporting

No comment

Experimental design

No comment

Validity of the findings

No comment

Additional comments

Dear authors,
I found this work of yours very successful. Your study plan has been created and implemented in detail. I just noticed that no sources were used in the first paragraph of the introduction. A source needs to be added to this section.

---

## Round 0.2 · accepted · Accept

I am pleased to accept this manuscript.